# Pelvic Pain in Reproductive Age: US Findings

**DOI:** 10.3390/diagnostics12040939

**Published:** 2022-04-09

**Authors:** Marco Di Serafino, Francesca Iacobellis, Maria Laura Schillirò, Francesco Verde, Dario Grimaldi, Giuseppina Dell’Aversano Orabona, Martina Caruso, Vittorio Sabatino, Chiara Rinaldo, Vito Cantisani, Gianfranco Vallone, Luigia Romano

**Affiliations:** 1Department of General and Emergency Radiology, “Antonio Cardarelli” Hospital, 80131 Naples, Italy; iacobellisf@gmail.com (F.I.); marialaura.schilliro@gmail.com (M.L.S.); francescoverde87@gmail.com (F.V.); dariogrimaldi@me.com (D.G.); giuseppinadellaversanoorabona@gmail.com (G.D.O.); caruso.martina90@gmail.com (M.C.); vittorio.sabatino@gmail.com (V.S.); chiara_rinaldo@libero.it (C.R.); luigia.romano1@gmail.com (L.R.); 2Department of Radiology, Sapienza Rome University, Policlinico Umberto I, 00185 Rome, Italy; vito.cantisani@uniroma1.it; 3Department of Life and Health, University of Molise “V. Tiberio”, 86100 Campobasso, Italy; gianfranco.vallone1@gmail.com

**Keywords:** pelvic pain, reproductive age, ultrasound, differential diagnosis

## Abstract

Pelvic pain in reproductive age often represents a diagnostic challenge due to the variety of potential causes characterized by overlapping clinical symptoms, including gynecological and other disorders (e.g., entero-colic or urological). It is also necessary to determine if there is a possibility of pregnancy to rule out any related complications, such as ectopic pregnancy. Although ultrasound (US), computed tomography (CT), and magnetic resonance imaging (MRI) are strongly integrated, the choice of which is the ideal diagnostic tool should be guided both by clinical suspicion (gynecological vs. non-gynecological cause) and by the risk ratio–benefit (ionizing radiation and instrumental costs), too. The didactic objective proposed by this review consists in the diagnosis of the cause and differential of pelvic pain in reproductive age by describing and critically analyzing the US diagnostic clues of the most frequent adnexal, uterine, and vascular causes.

## 1. Introduction

Pelvic pain in reproductive age often represents a diagnostic challenge due to the variety of potential causes characterized by overlapping clinical symptoms, including gynecological and other disorders (e.g., entero-colic or urological). It is also necessary to determine if there is a possibility of pregnancy to rule out any related complications, such as ectopic pregnancy [1]. Several imaging techniques can be adopted to image the female genital tract. The choice of the most suitable imaging approach and the related imaging protocols varies depending on the clinical indications and the patient conditions. Although ultrasound (US), computed tomography (CT), and magnetic resonance imaging (MRI) are strongly integrated, the choice of which is the ideal diagnostic tool should be guided both by clinical suspicion (gynecological vs non-gynecological cause) and by the risk–benefit ratio (ionizing radiation and instrumental costs), too [2]. Pelvic US is generally the first diagnostic tool used in this clinical setting and may require a transabdominal (TSA–US) and, if possible, transvaginal (TSV–US) approach to improve diagnostic sensitivity. CT and MRI may offer an added diagnostic value in pelvic pain assessment when US findings are unreliable or when pre-surgical evaluation is required [1]. The didactic objective proposed by this review consists in the diagnosis of the cause and differential of pelvic pain in reproductive age by describing and critically analyzing the US diagnostic clues of the most frequent adnexal, uterine, and vascular causes.

## 2. Examination Technique

Pelvic US is considered the ideal investigative tool to use at the beginning of the diagnostic and evaluation process for suspected gynaecological disorders in patients of all ages. US has the advantages of being widely available, low cost, and free from ionising radiation. Additionally, US is often sufficient to diagnose uterine, ovarian, and adnexal pathologies [1]. US is also extremely useful in evaluating pathological changes that can affect the pelvic portions of the urinary tract, the gastrointestinal tract, and in musculoskeletal structures, which can mimic the clinical picture of gynaecological pathology [1]. TSA–US and TSV–US are complementary techniques; both are used extensively in the evaluation of the female pelvis Table 1 [1].

However, the characteristics of pelvic pain must be strongly considered before any instrumental diagnostic approach. In this regard, Table 2 shows the most frequent adnexal, uterine, and vascular causes of pelvic pain in reproductive age where US has often a conclusive diagnostic role, also highlighting their clinical presentation characteristics.

## 3. Adnexal Causes of Pelvic Pain

### 3.1. Adnexal Torsion

Adenxal torsion is the partial or complete rotation of the adnexum on its vascular peduncle resulting in congestion and oedema, due to compromised venous and lymphatic drainage, and subsequent ischaemia and necrosis, due to compromise of the arterial blood supply [3,4]. Although frequently reported in the scientific literature, it is an uncommon cause of pelvic pain, accounting for only 3% of gynecologic emergencies [3].

It can affect the adnexum in its entirety involving both the ovary and the fallopian tube [4]. The torsion is often unilateral, and is more common on the right, presumably due to the reduced mobility of the left gonad due to the presence of the sigmoid colon. Torsion has a higher incidence in women of childbearing age and can be secondary to a space-occupying lesion, either cystic (large cystic lesions) or neoplastic, acting as a lead point [1,4,5]. Other potential causes include pregnancy, polycystic ovary syndrome, previous surgery to the pelvis, and hyperlaxity of the mesenteries and ovarian ligaments [1].

Since it is defined as a surgical emergency, an early differential diagnosis including other genitourinary and gastrointestinal conditions is essential to preserve the ovaries. The clinical symptoms are characterized by acute and sudden pain, sometimes associated with nausea and vomiting with the finding of a palpable pelvic mass and signs of peritonitis [1,3]. The initial assessment of a suspected adnexal torsion requires an US examination, which allows an increase in the volume of the ovary to be observed, which is generally greater than 4 cm, with or without an associated mass (Figure 1).

The ovary may be found in an abnormal position, close to the midline of the uterus, higher or lower in the pouch of Douglas, or displaced to the contralateral ovarian space [1,4,5]. The ovarian echotexture shows a hyperechoic central stroma due to the vascular congestion, associated with small peripheral follicles, with a “pearl necklace” appearance and, in the most advanced stages of vascular impairment, a diffusely uneven stroma due to the presence of focal haemorrhages and necrosis. The pathognomonic finding, however, remains the identification of the twisted vascular peduncle (“whirl” sign or “vascular vortex”), obtained by transversal scans along the central axis of the pedicle (Figure 2) [1,2,3,4,5].

The twisted and congested peduncle may appear as an indistinct adnexal mass adjacent to the twisted ovary. The colour Doppler examination allows for the assessment of the degree of torsion, the time elapsed from the beginning of the disease and therefore the degree of vascular compromise [1,2,3,4,5]. Vascular colour Doppler signals in the gonadal tissue are generally absent and this allows for a confident diagnosis with a positive predictive value of 94% (Figure 1).

However, the presence of flow does not exclude torsion, both due to the two-fold perfusion of the ovary by the ovarian and uterine arteries, despite the initial loss of venous flow, and because the torsion may be intermittent or incomplete. However, Doppler spectral waveform analysis can increase the sensitivity of the torsion diagnosis because an arterial waveform with diastolic flow inversion, indicating high resistance, may suggest the presence of torsion (Table 3) [1,2,3,4,5].

### 3.2. Ruptured or Bleeding Ovarian Cysts

Ovarian cysts are growths that develop in the ovaries during follicular maturation and are defined as functional cysts as they represent a physiological phenomenon related to ovarian function [1,6]. Functional ovarian cysts are caused by the overgrowth of a follicle due to the accumulation of fluid inside it, which is usually spontaneously reabsorbed without causing pain [6]. In some cases, they can rupture, releasing fluid into the peritoneal cavity, causing intense pain and bleeding complications. Haemorrhagic or ruptured ovarian cysts are common in women of reproductive age; however, the actual incidence is difficult to estimate, as many ruptured cysts are asymptomatic or found incidentally [7]. A broader estimate calculates about 7% of women worldwide experience a symptomatic cyst during their lifetime and in this event, US is the primary investigative tool [6,7,8]. The cyst may not be detectable if it has ruptured completely and decompressed; however, free fluid will generally be present in the pelvis. Therefore, rupture of ovarian cysts is often a diagnosis of exclusion when no other potential cause of the pain is identified [1,6,7,8]. The appearance of a haemorrhagic cyst detected by US is also variable and depends on the stage at which the patient has the scan. In the acute phase, it is possible to observe widespread homogeneous echoes with a fluid level and a dependent echogenic component represented by the blood sediment; over time, with the progressive degradation of haemoglobin, a reticular structure with internal echoes is observed (also described as a fishing net, spider web, or lace-like structure) (Figure 3) [1].

The fibrin bundles produce a network of precise linear echoes and are distinguishable from the real septa because they are numerous, thin, and dispersed, with an irregular structure and are not vascularised on colour Doppler US or contrast-enhanced US (CEUS) (Figure 4) [1].

In addition, unlike a true septum, they do not extend from one wall of the cyst to the other. The lack of flow detected by the colour Doppler imaging is an important feature, suggesting the presence of a mural thrombus rather than a tumour nodule. However, low levels of flow in solid tissue may not be detected by colour Doppler US, whereas contrast CEUS may optimise the diagnosis when clinical suspicion is present (Table 4) [1,6,7,8]. 

### 3.3. Pelvic Inflammatory Disease

Pelvic inflammatory disease (PID) is an infection of the female genital tract caused by the ascent of microorganisms from the vagina to the uterus, fallopian tubes, and ovaries. Common causes are Neisseria gonorrhoeae and Chlamydia trachomatis, which are sexually transmitted [1,9]. Other aerobic and anaerobic bacteria can cause bacterial vaginosis with retrograde spread of vaginal microorganisms [1]. The continuum of infection begins with cervicitis causing mucopurulent discharge and progresses to endometritis and salpingitis. Pus can collect in the tube (pyosalpinx) and form a tubo-ovarian abscess.

No specific international data are available for PID incidence worldwide. However, among 1171 sexually experienced reproductive-aged women in the 2013–2014 National Health and Nutrition Education Survey (NHANES), the prevalence of self-reported lifetime PID was 4.4% [10].

PID can be acute, chronic, or subclinical and is often underdiagnosed. Symptoms include lower abdominal pain, fever, mucopurulent discharge, and abnormal uterine bleeding during or after menstruation. Untreated PID can lead to chronic pelvic pain, infertility, ectopic pregnancy, and intra-abdominal infections. The diagnosis is made primarily on clinical suspicion [11]. In this setting, TSA–US of the abdomen and pelvis is frequently performed as the initial diagnostic imaging examination, to first rule out a broad spectrum of pathologic conditions, such as appendicitis, diverticulitis, adnexal torsion, and bowel obstruction [9,10,11].

On US, infected fallopian tubes may show thickened and hyperaemic walls, which are dilated in the presence of pyosalpinx with occluded ovarian fimbria, echogenic intraluminal sediment, and stratified echoes that signal the presence of exudate (Figure 5) [9].

The inflamed fallopian tube appears adjacent or adherent to the ovary with the formation, in the most advanced cases, of an ovarian abscess represented on US by an inflammatory mass which engulfs the ovary and the tube, making the ovary no longer distinguishable (Figure 6) [1,9].

Rupture of tubo-ovarian abscess can result in septic shock (Table 5) [1,9].

### 3.4. Endometritis

Endometritis is inflammation of the endometrial lining of the uterus that could also involve the myometrium and, occasionally, the parametrium too, and clinically manifest with fever, chills, lower abdominal pain, and foul-smelling lochia or PID related symptoms [12,13]. It can be divided into pregnancy-related endometritis with an incidence of 1–3% after a vaginal delivery and of 13–90% following cesarean delivery, and endometritis unrelated to pregnancy that may occur in up to 70–90% of documented cases of PID [12,13].

On US, inflammation of the endometrium can show a thickened endometrium with an irregular profile and the presence of more or less echogenic fluid or pus in the uterine cavity (pyometra) (Table 6) (Figure 7) [14].

### 3.5. Endometriosis

Endometriosis is a chronic inflammatory condition caused by the abnormal presence of endometrial tissue at sites other than the physiological endometrium. It affects up to 10% of women of reproductive age, is often asymptomatic and, when symptomatic, it manifests with chronic pelvic pain, dysmenorrhea, dyspareunia, and abnormal uterine bleeding [1,15,16,17,18]. Symptoms are often cyclical in nature as endometriosis is a hormone-responsive disease. Infertility is an important consequence of endometriosis, due to the anatomical deformation of the pelvic structures and occlusion of the fallopian tubes. Ectopic sites of endometriosis implantation include the ovarian surface, the suspensory ligaments of the uterus, the uterus itself, the peritoneal surfaces of the pouch of Douglas, and the fallopian tubes [1,15,16,17,18]. When the endometriosis tissue reaches structures deeper than 5 mm from the peritoneal surface causing fibrosis and muscle hyperplasia, it is defined as deep pelvic endometriosis. On US, endometriomas appear as unilocular swellings, which are often bilateral and multiple, with a thick capsule, regular margins, and homogeneously echogenic content, with fine internal echoes, due to the blood cells flaking off the walls, resulting in a “ground glass” appearance (Figure 8).

Endometriomas form part of the differential diagnosis alongside various ovarian cystic formations such as luteal, haemorrhagic, and dermoid cysts particularly. US guides the differential diagnosis, since, in the case of haemorrhagic cysts, monitoring in the different phases of the menstrual cycle allows for the observation of spontaneous resolution of the cysts, unlike endometriomas, which usually persist. In the presence of a non-definitive US examination or suspected ovarian cancer, an MRI scan is recommended [1,15,16,17,18]. Foci of endometriosis can also be detected on the anterior abdominal wall, the most common localisation of extrapelvic endometriosis, often next to a surgical or laparoscopic scar. High-frequency linear transducer ultrasound allows for the identification of a solid, heterogeneous, and hypoechoic mass with diffuse internal echoes (Figure 9) [1,15,16,17,18]. 

However, the characteristics detected are variable. The edges may be serrated and infiltrated by adjacent soft tissues. Most disease deposits show vascularisation on colour Doppler. The differential diagnosis includes abdominal wall masses, such as desmoid tumour, metastasis, lymphoma, melanoma, haematoma, suture granuloma, or incisional hernia; however, the location of the mass adjacent to a scar and a history of cyclic pain consistent with menstrual pain guide the diagnosis (Table 7) [1,15,16,17,18].

### 3.6. Peritoneal Inclusion Cyst

Peritoneal inclusion cysts (PICs) are generally benign mesothelial lesions with an estimate incidence of approximately 3–5% of the peritoneal mesotheliomas containing peritoneal fluid that occur in women of childbearing age, often following invasive pelvic surgery, infection, or cancer [19].

This condition is most commonly described in women of reproductive age with a ratio of 4–5:1 female to male, with very few cases reported in females over 30 years of age [19].

Intra-abdominal inflammation with reactive mesothelial proliferation and impaired absorption of peritoneal fluid can lead to its formation [20,21].

PICs are irregularly shaped, multiloculated adnexal cystic masses with angled margins, lacking a wall of their own and moulding themselves around the surrounding pelvic organs within the peritoneal cavity. A common clinical presentation is a progressive lower-abdominal and pelvic pain or a palpable mass [20,21]. Due to their rarity and non-specific clinical signs, the pre-operative diagnosis of PICs is challenging [20,21].

The *imaging* features of peritoneal inclusion cysts reflect their pathogenesis and allow for their differential diagnosis from an ovarian cystic mass [20,21]. Indeed, the typical US finding is an ovary trapped inside a cyst, surrounded by septa and fluid. The fluid is usually anechoic but may contain echoes in some compartments due to haemorrhage or protein-rich fluid (Figure 10).

Peritoneal inclusion cysts must also be differentiated from hydrosalpinx, which appears as a tubular or ovoid cystic formation, with visible folds, with no ovary inside of it (Table 8) [1,19,20,21].

### 3.7. Ectopic Pregnancy

The definition of ectopic pregnancy (EP) is that of a pregnancy that has implanted in a location other than the uterine cavity occurring in 1–2% of all pregnancies [22].

The most common abnormal implantation site for ectopic pregnancy is in the fallopian tubes, which occurs in up to 97% of cases. Of these, 75% to 80% are found in the ampullary region, 10% in the isthmic portion, 5% in the fimbrial portion, and 2 to 4% in the interstitial portion (Figure 11) [1,23,24,25,26]. 

Other less common sites include scarring of the ovary, abdomen, cervix, and uterus. The most common predisposing risk factor for EP is impaired tubal function, usually caused by tubal scarring, secondary to salpingitis associated with pelvic inflammatory disease triggered by sexually transmitted infections [23,24,25,26]. Additionally, the incidence of EP increases in women who have conceived through assisted reproduction techniques or in the presence of an intrauterine device. The joint detection of human chorionic gonadotropin (β-hCG) in the serum and an ultrasound scan, preferentially transvaginal, allows the physician to differentiate between intrauterine pregnancy and EP. Ectopic pregnancy should be suspected in patients with vaginal bleeding or lower abdominal pain and a positive pregnancy test in the absence of an intrauterine gestational sac visible on US [27].

In general, an intrauterine gestational sac is expected to be visualised when the b-hCG is 1000 mIU/mL (according to the international standard) or 2000 mIU/mL (according to the International Reference Preparation, IRP). When the b-hCG value is below the cut-off value of 2000 mIU/mL (IRP) and there is no intrauterine gestational sac, the diagnosis could be an early intrauterine pregnancy, a miscarriage, or an ectopic pregnancy, and therefore follow-up is indicated [23,24,25,26]. On the other hand, when the b-hCG value is above the cut-off value, you can expect to observe an intrauterine gestational sac which, if not visualised, will raise the suspicion of an indeterminate pregnancy [1,23,24,25,26]. However, US findings relating to the uterus are not specific and include the absence of an intrauterine gestational sac but with the finding of fluid in the endometrial canal. The latter, often containing low-level echoes, centrally located within the uterine cavity, and surrounded by a single echogenic decidual layer, is called a “pseudogestational sac” or “pseudosac”. It differs from a genuine intrauterine gestational sac, which can be either in an eccentric position or incorporated in a single layer of the endometrium (intradecidual sac) or at least partially surrounded by two echogenic tissue furrows with an interposed hypoechoic layer (double decidual sac). However, the most specific ultrasound finding for the diagnosis of EP is the presence of an ectopic gestational sac containing the yolk sac or embryo (with or without cardiac activity), while the most common consists of an echogenic tubal ring in the adnexa (Figure 12) (Table 9) [23,24,25,26].

## 4. Uterine Causes of Pelvic Pain

### 4.1. Fibroids: Degeneration, Rupture and Torsion

Fibroids are the most common benign uterine tumours found in women of childbearing age. Although usually asymptomatic, they can manifest with acute or chronic pelvic pain, particularly due to compressive effects on neighbouring organs. Given the high prevalence of fibroids, it is remarkable that acute complications are very rare indeed [28]. A study examining the incidence of degeneration of leiomyoma in patients referred for uterine fibroid embolisation underwent MRI found an incidence of 5.1% [29]. Torsion and rupture are a rare entity (reported incidence for torsion of less than 0.25%) [30].

An important clinical sign is abnormal uterine bleeding (Figure 13).

Their severity varies according to size, location, and number [31,32]. They can cause acute pelvic pain during pregnancy or in the postpartum period following significant hormonal stimulation and an acute infarction. Rupture of a degenerated fibroid is a rare complication and can be complicated by an acute, potentially life-threatening abdominal haemoperitoneum [19,20]. When pedunculated, fibroids can undergo torsion and cause acute painful symptoms. On US examination, the torsion of a fibroid appears as a hypovascular or avascular mass distinct from the ovary and with a twisted or “pointed” peduncle (Table 10) [1,31,32].

### 4.2. Post-Embolisation Syndrome

Uterine artery embolisation for the treatment of fibroids is indicated in symptomatic women as an alternative to hysterectomy. Post-embolisation syndrome, which occurs in about 40% of women undergoing this procedure, is characterised by fever, pelvic pain, and vaginal bleeding, which subsides within 24 to 48 h [33]. The repeat US allows for the observation of changes in the appearance of the fibroid. The physician may note the presence of air inside the fibroid, which is represented on US with internal echoes and reverberation artifacts in the context of a poorly defined mass following infarction and tissue necrosis (Table 11) (Figure 14) [1,33].

This finding, if isolated, is not indicative of infection. However, the correlation between the patient’s symptoms and clinical findings is essential to promptly diagnose superinfection and, consequently, to assess the possible development of a pyomyoma (suppurative fibroid) with a high risk of septic shock [33]. CT may be performed as a complement to diagnosis to evaluate the possible presence of associated complications, such as pelvic abscess, rupture of the uterus, or septic thrombophlebitis of the ovarian vein [1,33].

## 5. Vascular Causes of Pelvic Pain

### 5.1. Pelvic Congestion Syndrome

Pelvic congestion syndrome (PCS) is characterized by chronic symptoms that may include pelvic pain, perineal heaviness, urinary urgency, and postcoital pain, caused by valvular insufficiency of the ovarian veins, resulting in reflux to the pelvic veins and vulvar, perineal, and lower limb varices. In patients with presenting complaints of chronic pelvic pain, the prevalence of PCS is nearly 30% [34,35].

It is estimated that not all patients with pelvic varicose veins have pain and that approximately 40–60% of women with pelvic varicose veins and reflux develop pelvic congestion syndrome [1,36]. According to the International Union of Phlebology consensus document, the dilation of pelvic veins is defined as an increase in their diameter of greater than 5 mm, and the pelvic venous reflux is considered pathological if it lasts for greater than 1 s [37].

Predisposing factors for the disease are age between 20 and 40 years, retroverted uterus, multiparity and pelvic surgery [1]. Affected women complain of pain after prolonged standing, lifting weights, coitus, or during the premenstrual period. Low BMI is a risk factor for pelvic congestion syndrome [38].

US examination permits to exclude pelvic masses, cystic changes in the ovaries, and uterine pathologies as potential causes of pain and represents the first line diagnostic test to evaluate pelvic congestion syndrome [37,39]. Indeed, on US examination, it is possible to observe multiple veins with a diameter greater than 5 mm adjacent to the ovary and uterus and enlarged arcuate veins also with a diameter greater than 5 mm, which can cross the myometrium and connect to varicosities (Figure 15) [1,36].

However, the diameter of pelvic veins is not a diagnostic criterion to differentiate symptomatic and asymptomatic PCS, how much more the duration of pelvic venous reflux is greater than 1 second, its prevalence in the pelvic veins, and blood deposition in the pelvic venous plexuses, including the uterine and parametrial veins and not only the ovarian ones, are usually the leading factors in the development of symptomatic forms of pelvic congestion (Table 12) [40,41].

In this regard, it appears critical to have a rigorous methodological US approach of pelvic and retroperitoneal veins combining TSA–US and TVS–US with dynamic colour-Doppler Valsalva maneuvers performed in the patient’s supine, half-sitting (with trunk raised to 45°), and half-standing positions [40,41]. TVS–US with colour-Doppler is considered as the gold standard investigation for the hemodynamic assessment of pelvic veins reflux in women since it offers better visualization of the pelvic venous plexus compared to TSA–US and is not hampered by patient habitués or undisplaceable bowel gas [37,40,41,42]. TSA–US with colour Doppler of the iliac veins, inferior vena cava, renal, and gonadal is useful in searching functional causes of pelvic reflux such as incompetent gonadal vein valves or structural causes of pelvic reflux such as renal vein, iliac vein, and/or inferior vena cava compression and/or abnormalities, too [40,41,42].

Finally, duplex-ultrasonography of the veins of lower extremities is also a necessary part of the imaging protocol for improved evaluation of pelvic congestion syndrome, especially in the presence of atypical varicose veins [37].

### 5.2. Thrombosis of the Gonadal Veins

Gonadal vein thrombosis is a condition that can occur in postpartum women or those undergoing pelvic surgery, with a referred incidence of about 0.18% of the general population [43,44]. Affected patients present with acute pain, often with fever and leucocytosis on laboratory examination. Thrombosis is observed more frequently on the right, probably due to the greater pressure present in the right gonadal vein than on the left, where the pressure would be protected by the retrograde flow of the left renal vein; the right ovary, the ipsilateral iliopsoas muscle, and the inferior vena cava are therefore frequently affected [1,43]. Uterine venous plexus thrombosis is also an unusual site of thrombosis often asymptomatic and incidental detected by TSV–US, which is decisive in the diagnosis and subsequent therapeutic choices [45]. US approach of gonadic vein thrombosis combines TSA–US and TSV–US. The latter is referred to the US method to explore the intere gonadic vein decourse and it is performed through transverse and longitudinal scans of the retroperitoneum showing an avascular structure with a tortuous tubular appearance, with adjacent anechoic or hypoechoic areas without any flow detection on the colour Doppler evaluation (Table 13) [46].

CT or MRI, both with contrast medium, should be carried out in the event of an uncertain diagnosis and to assess the extent of thrombosis (Figure 16). Furthermore, Time-of-flight (TOF) sequence is a non-invasive MRI technique that can be used to visualize thrombus as filling defects within gonadal veins, without the need to administer contrast [47].

## 6. Conclusions

Pelvic US is the ideal investigative tool to adopt as first line examination for suspected gynaecological disorders in patients of all ages. As pictured, US allows for an accurate diagnosis of ovarian torsion, hemorrhagic ovarian cyst, endometriosis, pelvic inflammatory disease, complications related to leiomyomas, and vascular anomalies first by placing them in the differential diagnosis and between other non-gynecological causes of pelvic pain such as gastrointestinal and genitourinary disorders. The recognition of such disorders can make timely diagnosis possible and avoid further imaging tests.

## Figures and Tables

**Figure 1 diagnostics-12-00939-f001:**
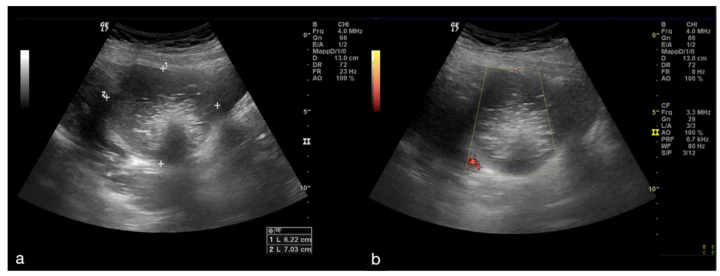
Adnexal torsion of giant mature cystic right teratoma. Axial TSA-US image (**a**) shows a mostly echogenic mass (caliper) with some sound attenuation. On power Doppler (**b**) no adnexal vascularization is detected.

**Figure 2 diagnostics-12-00939-f002:**
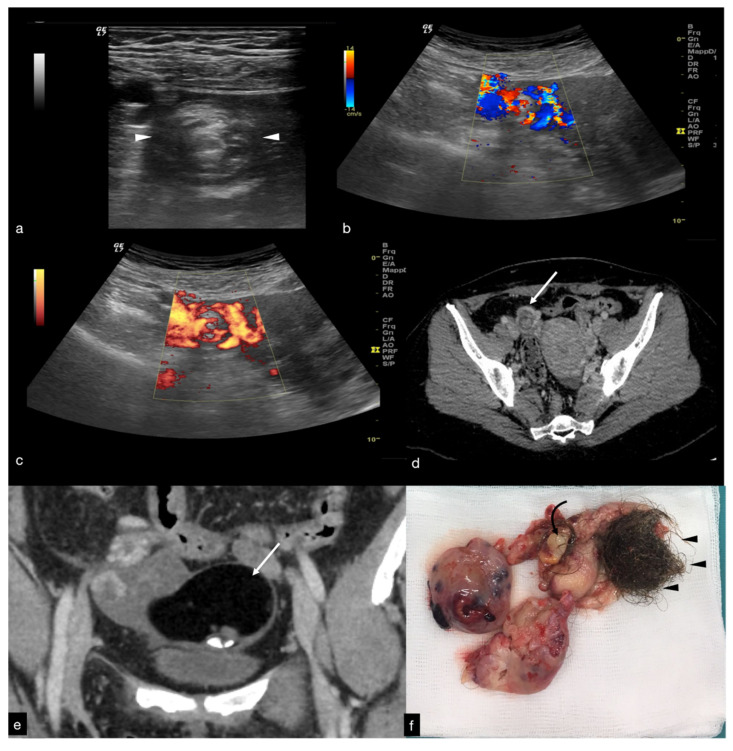
Same patient of Figure 1. Axial TSA–US image with high frequency probe (**a**) shows a twisted vascular peduncle (“whirl” sign) from the side of the right adnexal mass of above Figure 1. The twisted adnexal peduncle is better seen at colour-Doppler (**b**) and power-Doppler evaluation (**c**) and confirmed also at enhanced CT ((**d**) arrow), related with the presence of a large mature cystic teratoma ((**e**) arrow). At the operative specimen, ovarian torsion is confirmed, and the right ovary with the cystic teratoma with its mixed content of hairs ((**f**) arrowheads) and teeth ((**f**) curved arrow) is shown. Reprinted with permission from [2].

**Figure 3 diagnostics-12-00939-f003:**
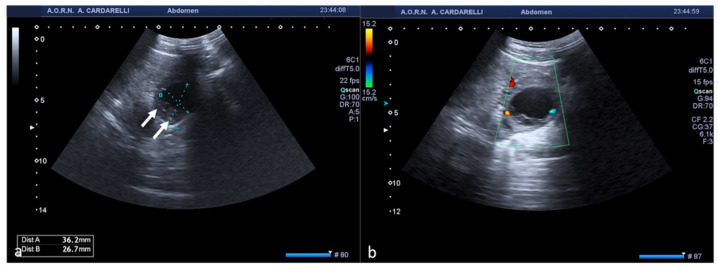
Haemorrhagic left ovarian cyst. Axial TSA-US image (**a**) shows a hemorrhagic corpus luteum ((**a**) caliper) with blood deposits ((**a**) arrows). On colour-Doppler imaging (**b**) circumferential blood flow is also shown.

**Figure 4 diagnostics-12-00939-f004:**
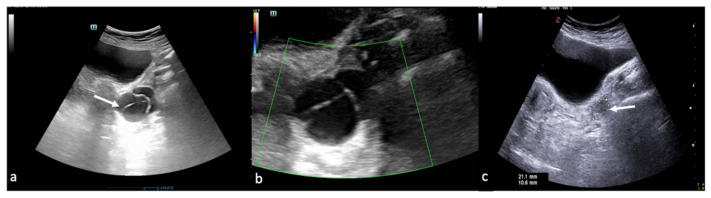
Haemorrhagic left ovarian cyst. Axial US image (**a**) shows a hemorrhagic cyst ((**a**) arrow) with reticular structure. On colour-Doppler imaging (**b**) there is no blood flow of the reticular structure. At follow-up (**c**) a complete resorption of the cyst is observed ((**c**) arrow).

**Figure 5 diagnostics-12-00939-f005:**
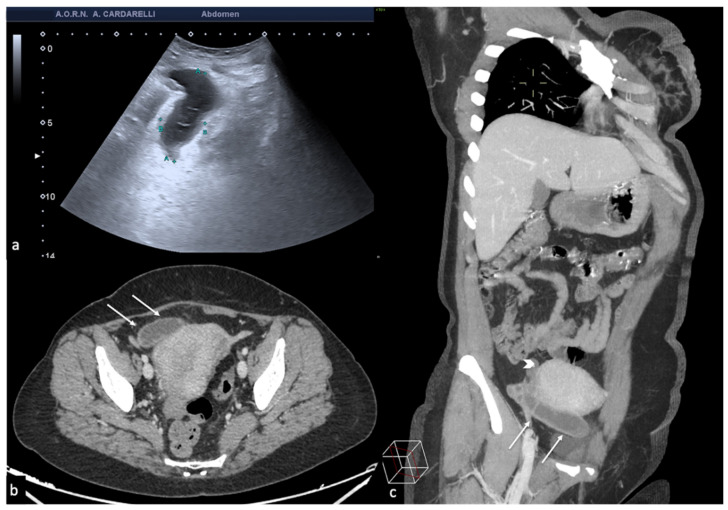
Pelvic inflammatory disease. US scan of the right adnexa (**a**) shows an anechoic oblong structure ((**a**), caliper). Contrast-enhanced axial CT image (**b**) and maximum intensity projection (MIP) coronal-oblique reconstruction (**c**) show tortuous and tubular fluid filled structure seen in the right adnexa ((**b**,**c**) arrows).

**Figure 6 diagnostics-12-00939-f006:**
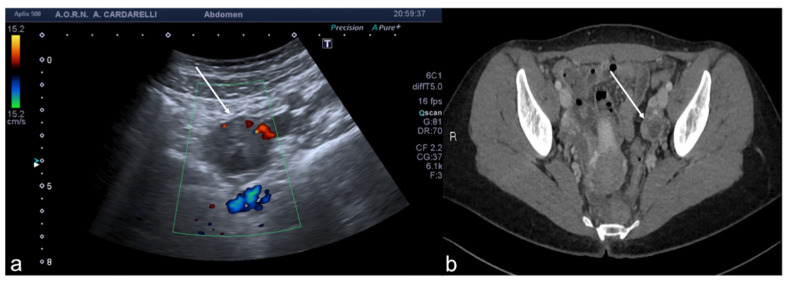
Tubo-ovarian complex. Axial colour-Doppler TSA–US scan of the left adnexa (**a**) shows a complex cystic mass with peripheral signal colour Doppler ((**a**) arrow). Contrast-enhanced axial CT image (**b**) shows a thick enhancing left adnexa abscess wall ((**b**) arrows).

**Figure 7 diagnostics-12-00939-f007:**
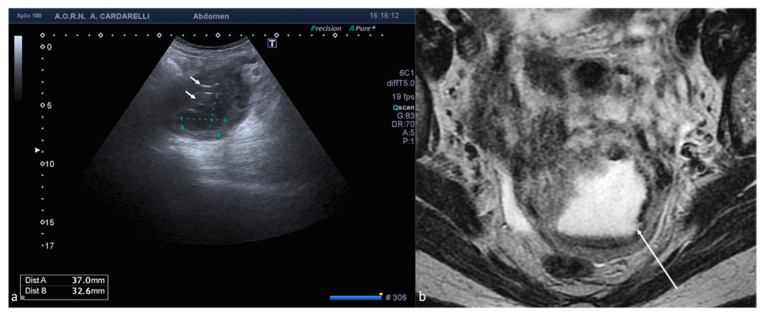
Pyometra. TSA-US scan of the uterus (**a**) shows a distended uterine cavity containing complex fluid with echogenic foci (short arrows). Axial T2w MRI imaging (**b**) confirms the diagnosis showing pus collection into uterine cavity.

**Figure 8 diagnostics-12-00939-f008:**
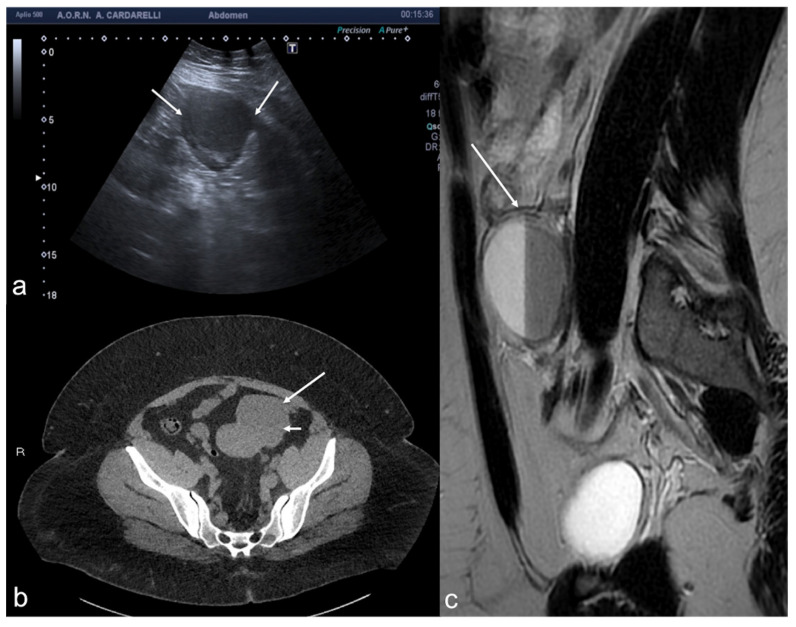
Adnexal endometriomas. TSA–US axial scan (**a**) shows a left adnexal unilocular cyst with acoustic enhancement and diffuse homogeneous ground-glass echoes as a result of the hemorrhagic debris (arrows). On axial unenhanced CT scan, a hypodense mass in the left annex ((**b**) long arrow). Note the fluid-thick level ((**b**) short arrow). This latter finding appears clearly visible on T2w MRI ((**c**) arrow).

**Figure 9 diagnostics-12-00939-f009:**
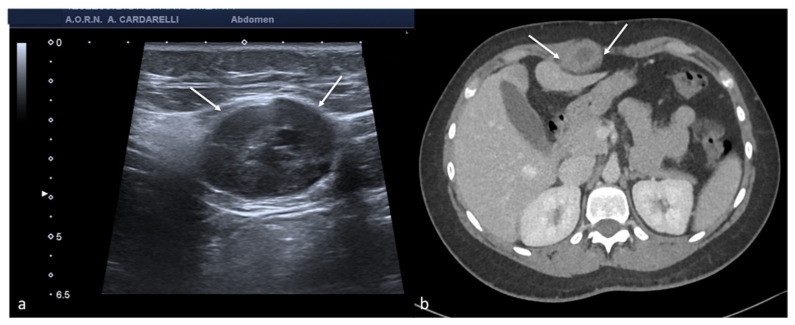
Abdominal wall endometriosis nodule. TSA–US axial scan (**a**) of the abdominal wall shows a 35 mm abdominal wall endometriosis nodule with hypoechoic content and well-defined margins ((**a**), arrows). The nodule is enclosed in the muscular fascia along the right rectal abdomen. Contrast-enhanced axial CT image (**b**) shows heterogeneous enhanced mass in the right rectus sheath. The mass was subsequently proved to be abdominal endometriosis.

**Figure 10 diagnostics-12-00939-f010:**
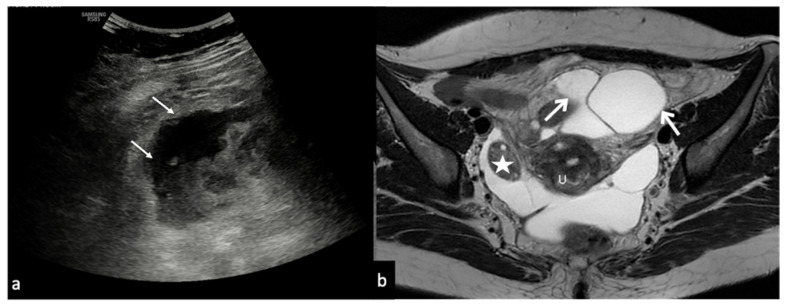
Peritoneal inclusion cyst. TSA–US axial scan (**a**) shows an irregular, anechoic para-adnexal right fluid collection ((**a**) arrows). Axial T2w MRI (**b**) shows a large, well-defined multilocular cystic pelvic mass (arrows). Note several thin internal septa, radiating from the ovary, representing pelvic adhesions. The septa radiate from the ovary (star; (**b**)). U: uterus.

**Figure 11 diagnostics-12-00939-f011:**
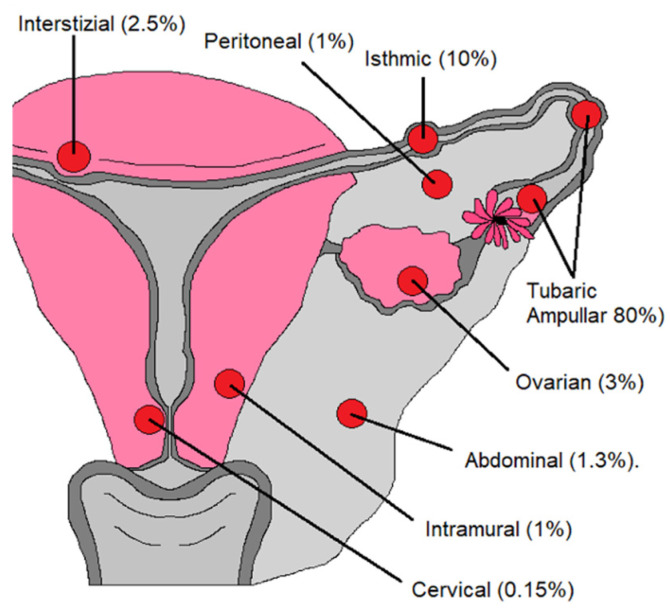
Abnormal implantation sites for ectopic pregnancy.

**Figure 12 diagnostics-12-00939-f012:**
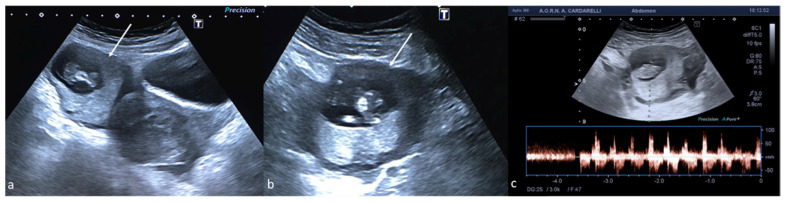
Ectopic pregnancy. Axial (**a**) and longitudinal (**b**) TSA–US scans reveal an extrauterine adnexal gestational sac with a fetal body ((**a**,**b**), arrow). On pulsed Doppler (**c**) a heartbeat is also detected.

**Figure 13 diagnostics-12-00939-f013:**
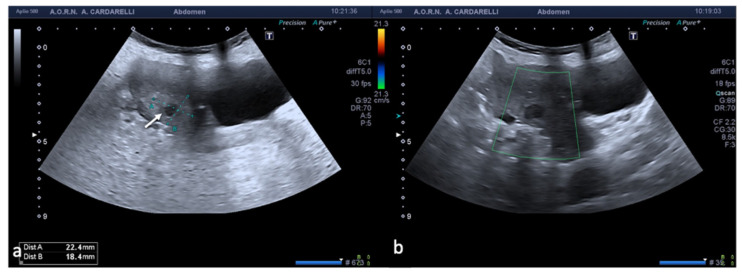
Fibroid degeneration. Axial (**a**) TSA–US scan shows a heterogeneous uterine mass ((**a**), caliper) with cystic areas inside ((**a**) arrow). On colour Doppler (**b**) no vascular signal is detected.

**Figure 14 diagnostics-12-00939-f014:**
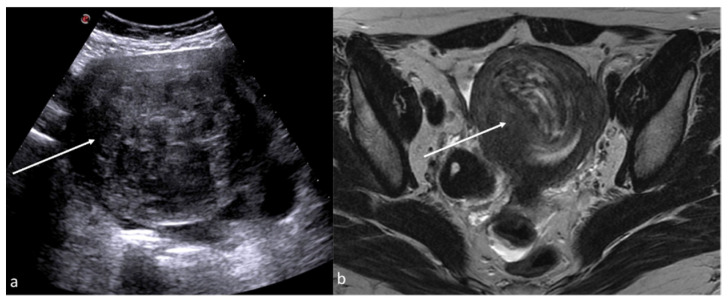
Post-embolization fibroid necrosis. Axial TSA-US (**a**) scan shows a heterogeneous fibroid ((**a**) arrow) in necrotic evolution. Axial T2w MRI (**b**) confirmed the diagnosis (arrow).

**Figure 15 diagnostics-12-00939-f015:**
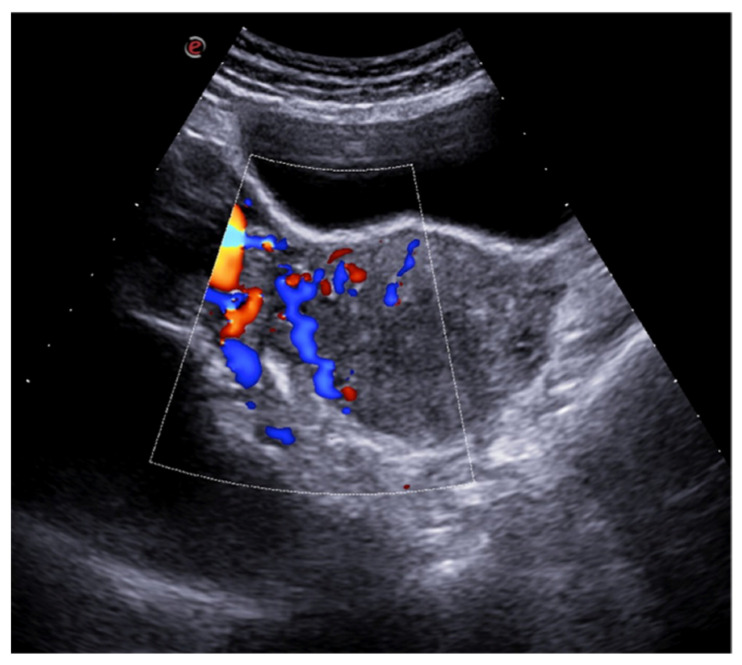
Pelvic congestion syndrome. US axial scan with colour-Doppler mode shows dilated veins in the right adnexa with reversed venous flow after Valsalva maneuver.

**Figure 16 diagnostics-12-00939-f016:**
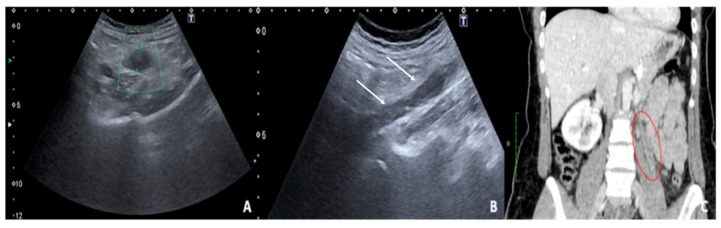
Thrombosis of the right gonadal vein. TSA–US scan of the left ovarian vein in axial (**A**) and longitudinal (**B**) view before delivery show the ovarian vein as a tubular structure with heterogeneous hypoechoic echotexture ((**B**) arrows), located superiorly to the ovary and anteriorly to the psoas muscle. Contrast-enhanced CT coronal image (**C**) was performed after delivery and confirmed the left gonadic vein thrombosis (red circle). Reprinted with permission from [46].

**Table 1 diagnostics-12-00939-t001:** TSA–US and TSV–US in comparison.

US Methods	Probe	Protocol	Utility	Limits
Transabdominal sonography (TSA–US)	Low-frequency probe (convex probe 3–5 MHz).	The standard protocol for examining the female pelvis involves an initial TAS with the urinary bladder completely full, so it can act as an acoustic window. Following bladder emptying, the patient assumes a lithotomy position, and TVS is performed. The two imaging techniques are complementary and often provide different diagnostic information. The protocol often also calls for the execution of Doppler, power Doppler, and pulsed wave Doppler flowmetry depending on the clinical situation and pathology that emerge from grayscale imaging.	TAS offers a wider field of view than TVS and allows better visualisation of the superficial and distal structures of the vagina by bringing the probe closer to the target organs.	Empty bladder. Obese patients. Retroverse uterus where the fundus is located beyond the focal zone of the transducer. Less effective for characterisation of adnexal masses.
Transvaginal sonography (TVS–US)	High-frequency probe (endocavitary probe > 7 MHz).		TVS approach requires a greater penetration depth to avoid the attenuating soft tissues that cover the pelvic organs. Therefore, it requires the use of a higher frequency probe, which, in turn, provides greater resolution of the anatomical details of the uterus, ovary, and adnexal structures.	Limited field of view. Should not be performed on patients who are unable or unwilling to consent to the procedure, as well as on most virgin patients and for those in which the insertion of the probe produces marked discomfort. It is contraindicated in some obstetric patients in the 2nd and 3rd trimester of pregnancy due to the risk of active bleeding or membrane rupture.

**Table 2 diagnostics-12-00939-t002:** Pelvic pain in reproductive age: causes, symptoms, and occurrence.

Causes of Pelvic Pain	Occurrence	Pain Characteristics
**• Adnexal**		
Adnexal torsion	3% of gynecologic emergencies	Acute persistent (complete torsion) or intermittent (intermittent torsion) right/left pelvic pain
Ruptured or bleeding ovarian cysts	The incidence is difficult to estimate. A broader estimate calculates about 7% of women worldwide experience a symptomatic cyst during their lifetime	Acute right/left pelvic pain
Pelvic inflammatory disease	No specific international data are available for PID incidence worldwide. A study reports a prevalence of self-reported lifetime PID of 4.4%	Chronic pelvic pain with reacutization
Endometritis	Pregnancy-related endometritis with an incidence of 1–3% after a vaginal delivery and of 13–90% following cesarean delivery, and endometritis unrelated to pregnancy that may occur in up to 70–90% of documented cases of PID	Chronic pelvic pain with reacutization
Endometriosis	It affects up to 10% of women of reproductive age	Asymptomatic/poorly symptomatic/chronic pelvic pain with reacutization during menses
Peritoneal inclusion cysts	Approximately 3–5% occur in women of childbearing age following invasive pelvic surgery, infection, or cancer	Asymptomatic/poorly symptomatic/chronic pelvic pain with reacutization (especially if complicated)
Ectopic pregnancy	1–2% of all pregnancies	Acute pelvic pain
**• Uterine**		
Fibroids: degeneration, rupture, and torsion	A study examining the incidence of degeneration of leiomyoma in patients referred for uterine fibroid embolisation underwent MRI found an incidence of 5.1%. Torsion and rupture are a rare entity (reported incidence for torsion of less than 0.25%)	Acute pelvic pain
Post-embolisation syndrome	Occurs in about 40% of women undergoing uterine artery embolisation	Pelvic pain of variable entity
**• Vascular**		
Pelvic congestion syndrome	In patients with presenting complaints of chronic pelvic pain, the prevalence of PCS is nearly 30%	Chronic pelvic pain
Thrombosis of the gonadal veins	Referred incidence of about 0.18% of the general population	Acute pelvic pain of variable entity

**Table 3 diagnostics-12-00939-t003:** Adnexal torsion US diagnostic clue.

Adnexal Causes of Pelvic Pain	US Diagnostic Clue	US Limits
Adnexal torsion	Twisted vascular peduncle (whirl sign) with absent flows or with increase in resistance indices if represented	Transabdominal US may be limited in obese patients or when the ovaries are masked by intestinal meteorism. Endovaginal US may be limited in cases of large ovarian masses causing cranial displacement of the ovary, hindering the exploration of the ovarian vessels.

**Table 4 diagnostics-12-00939-t004:** Ruptured or bleeding ovarian cysts US diagnostic clue.

Adnexal Causes of Pelvic Pain	US Diagnostic Clue	US Limits
Ruptured or bleeding ovarian cysts	Cystic mass with an inhomogeneous echo structure in relation to hemoglobin degradation often with evidence of haematic sediment or in an advanced phase with relief of thin internal echoes arranged in a “fishing net” or fibrin bundles not vascularized by colour-Doppler or CEUS (differential diagnosis with tumor mass).	US cannot detect and quantify the active bleeding

**Table 5 diagnostics-12-00939-t005:** Inflammatory disease US diagnostic clue.

Adnexal Causes of Pelvic Pain	US Diagnostic Clue	US Limits
Pelvic inflammatory disease	The fallopian tubes with thickened and hyperemic walls, dilated in the presence of pyosalpinx with occluded ovarian fimbria, echogenic intraluminal sediments, and echoes stratified by exudate. The inflamed fallopian tube appears adjacent to or adhering to the ovary with the formation, in more advanced cases, of an ovarian tube abscess represented on US by an inflammatory mass that engulfs the ovary and the fallopian tube, no longer making the ovary distinguishable.	US may suffer from limited panoramicity and, in cases of extensive adhesions, can be difficult to discriminate each anatomical structure.

**Table 6 diagnostics-12-00939-t006:** Endometritis US diagnostic clue.

Adnexal Causes of Pelvic Pain	US Diagnostic Clue	US Limits
Endometritis	Thickened endometrium with an irregular profile and the presence of more or less echogenic fluid or pus in the uterine cavity (pyometra)	At US, may be difficult to distinguish severe endometritis from cancer

**Table 7 diagnostics-12-00939-t007:** Endometriosis US diagnostic clue.

Adnexal Causes of Pelvic Pain	US Diagnostic Clue	US Limits
Endometriosis	Unilocular swellings, which are often bilateral and multiple, with a thick capsule, regular margins, and homogeneously echogenic content, with fine internal echoes, due to the blood cells flaking off the walls, resulting in a “ground glass” appearance. Useful monitoring for differential diagnosis with hemorrhagic ovarian cyst (persistence to follow-up of the endometriotic cyst addresses the diagnosis)	At US, may be difficult to detect millimetric foci of ovarian endometriosis and to detect retrocervical or ligaments thickening, as well as possible intestinal or nerve involvement.

**Table 8 diagnostics-12-00939-t008:** Peritoneal inclusion cyst US diagnostic clue.

Adnexal Causes of Pelvic Pain	US Diagnostic Clue	US Limits
Peritoneal inclusion cyst	Trapped ovary inside a cyst, surrounded by septa and fluid. The fluid is usually anechoic but may contain echoes in some compartments due to haemorrhage or protein-rich fluid	At US, can be difficult to differentiate the cystic origin.

**Table 9 diagnostics-12-00939-t009:** Ectopic pregnancy US diagnostic clue.

Adnexal Causes of Pelvic Pain	US Diagnostic Clue	US Limits
Ectopic pregnancy	When the b-hCG value is below the cut-off value of 2000 mIU/mL (IRP) and there is no intrauterine gestational sac, the diagnosis could be an early intrauterine pregnancy, a miscarriage, or an ectopic pregnancy, and therefore follow-up is indicated Uterine findings: gestational pseudosac (differential diagnosis with gestational sac: double echogenic wall versus single echogenic wall of the pseudosac). Adnexal findings: echogenic tubal ring or ectopic gestational sac containing the yolk sac or the embryo (with or without cardiac activity)	Related with difficulties in exploring the adnexa and in detecting early pregnancy as well as active bleeding in case of rupture

**Table 10 diagnostics-12-00939-t010:** Fibroid torsion US diagnostic clue.

Uterine Causes of Pelvic Pain	US Diagnostic Clue	US Limits
Fibroids torsion	Hypovascular or avascular mass distinct from the ovary and with a twisted or “pointed” peduncle	Difficulties in exploring the whole uterus in cases of multiple fibromas and in detecting the twisted pedicle

**Table 11 diagnostics-12-00939-t011:** Post-embolisation syndrome US diagnostic clue.

Uterine Causes of Pelvic Pain	US Diagnostic Clue	US Limits
Post-embolisation syndrome	Fibroid with internal echoes and reverberation artifacts in the context of a poorly defined mass following infarction and tissue necrosis	Difficulties in exploring the whole uterus in cases of multiple fibromas

**Table 12 diagnostics-12-00939-t012:** Pelvic congestion syndrome US diagnostic clue.

Vascular Causes of Pelvic Pain	US Diagnostic Clue	US Limits
Pelvic congestion syndrome	Multiple pelvic veins with a diameter greater than 5 mm and a venous reflux greater than 1 s	Possible difficulties in sampling the vessel and correctly evaluate the blood flow, and in detecting eventual complications such as thrombosis.

**Table 13 diagnostics-12-00939-t013:** Thrombosis of the gonadal veins US diagnostic clue.

Vascular Causes of Pelvic Pain	US Diagnostic Clue	US Limits
Thrombosis of the gonadal veins	Avascular structure with a tortuous tubular appearance, with adjacent anechoic or hypoechoic areas without any flow detection on the colour Doppler evaluation.	Possible limits in panoramicity that hinder the detection of thrombosed vessels. Difficulties in Doppler evaluations.

## Data Availability

Data sharing is not applicable.

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
