# Peer review of "Pelvic Pain in Reproductive Age: US Findings"

_diagnostics, 2022, doi:10.3390/diagnostics12040939_

Round 1

Reviewer 1 Report

Thank you for submitting important paper “Pelvic pain in reproductive age: US findings”

Here is minor comments on your paper.

Line 333

Predisposing factors for the disease are age between 20 and 333 40 years, retroverted uterus, multiparity and pelvic surgery [1]. Affected women complain of pain after prolonged standing, lifting weights, or during the premenstrual period

Please add coitus after “prolonged standing, lifting weights,”

Low BMI is risk factor of Pelvic congestion.

“Nanavati R. Correlation between pelvic congestion syndrome and body mass index. J Vasc Surg. 2018 Feb;67(2):536-541. “

Line 338

On US examination, it is possible to observe multiple veins with a diameter 338

How about add US examination is the first line diagnostic test as stated in “Borghi C, Dell'Atti L. Pelvic congestion syndrome: the current state of the literature. Arch Gynecol Obstet. 2016 Feb;293(2):291-301. doi: 10.1007/s00404-015-3895-7. Epub 2015 Sep 24. PMID: 26404449.”

Line 355

on the colour Doppler evaluation. CT or MRI, both with contrast medium, should be

Dose MRI also needs contrast medium?

Author Response

Thank you for submitting important paper “Pelvic pain in reproductive age: US findings”

Thank you for your comments helpful to improve manuscript quality

Here is minor comments on your paper.

Line 333: Predisposing factors for the disease are age between 20 and 40 years, retroverted uterus, multiparity and pelvic surgery [1]. Affected women complain of pain after prolonged standing, lifting weights, or during the premenstrual perioD. Please add coitus after “prolonged standing, lifting weights,”

We have added “coitus" in the list. Affected women complain of pain after prolonged standing, lifting weights, coitus, or during the premenstrual period. Low BMI is a risk factor for pelvic congestion syndrome [38].

Low BMI is risk factor of Pelvic congestion. “Nanavati R. Correlation between pelvic congestion syndrome and body mass index. J Vasc Surg. 2018 Feb;67(2):536-541. “

We have added the risk factor “low BMI” with the related reference.  Low BMI is a risk factor for pelvic congestion syndrome [38].

Line 338 On US examination, it is possible to observe multiple veins with a diameter   How about add US examination is the first line diagnostic test as stated in “Borghi C, Dell'Atti L. Pelvic congestion syndrome: the current state of the literature. Arch Gynecol Obstet. 2016 Feb;293(2):291-301. doi: 10.1007/s00404-015-3895-7. Epub 2015 Sep 24. PMID: 26404449.”

We have added this reference and highlight the role of US in the evaluation of pelvic congestion syndrome. US examination permits to exclude pelvic masses, cystic changes in the ovaries and uterine pathologies as potential causes of pain and represents the first line diagnostic test to evaluate pelvic congestion syndrome [37, 39].

Line 355 on the colour Doppler evaluation. CT or MRI, both with contrast medium, should be. Dose MRI also needs contrast medium?

We have added a sentence about the MRA without the need to administer contrast, using TOF sequence with the related reference. Furthermore, Time-of-flight (TOF) sequence is a non-invasive MRI technique that can be used to visualize thrombus as filling defects within gonadal veins, without the need to administer contrast [47].

Reviewer 2 Report

I am grateful to the editors of the Diagnostics journal for the opportunity to review the manuscript entitled: "Pelvic pain in reproductive age: US findings." The use of ultrasonography for identifying causes of pelvic pain is among the crucial issues not only in obstetrics and gynaecology, but also in vascular surgery, urology, and traumatology. In this regard, an important point is the standardization of the estimated parameters of abnormalities of pelvic organs and veins and retroperitoneal space. Unfortunately, an ultrasonographer is usually specialised in one area of medicine (e.g., in gynaecology or surgery), which leads to underdiagnosis of diseases outside his/her specialization. And underdiagnosis of pelvic venous disease in patients with pelvic pain is not such a rare example of that.

 The presented manuscript confirms this thesis to a certain extent. Having chosen the most frequent organ causes of pelvic pain, the authors forgot about the methodological features of the study on the pelvic vessels (arteries and veins), while the correct interpretation of changes in pelvic veins can significantly increase the diagnostic value of ultrasonography in patients with pelvic pain. One should not forget about visceral artery aneurysms, post-thrombotic lesions of the iliac veins.

Comments

  1. Regarding the title, it is not clear why the authors focused on only the reproductive age. Pelvic pain may develop at any age. Moreover, the risks and incidences of diseases associated with pelvic pain in women increase with aging. In my opinion, the emphasis on age should be removed from the title.

  1. Examination technique. I fully admit that the standard protocol for examining the pelvic organs does not require a detailed description. However, with regard to the examination of pelvic veins, it is necessary to describe features of the study of pelvic and retroperitoneal veins: the position of the patient (there are several of them), stress tests, veins classified as pelvic, the study of main venous trunks (left renal vein, iliac veins), and reflux. In this regard, the authors should get acquainted with the following articles and cite them in the References: Whiteley MS, Dos Santos SJ, Harrison CC, Holdstock JM, Lopez AJ. Transvaginal duplex ultrasonography appears to be the gold standard investigation for the haemodynamic evaluation of pelvic venous reflux in the ovarian and internal iliac veins in women. Phlebology. 2015;30(10):706-713; Gavrilov S, Karalkin A, Mishakina N, Efremova O, Grishenkova A. Relationships of Pelvic Vein Diameter and Reflux with Clinical Manifestations of Pelvic Venous Disorder. Diagnostics (Basel). 2022;12(1):145; Gavrilov S, Moskalenko YP, Mishakina NY, Efremova OI, Kulikov VM, Grishenkova AS. Stratification of pelvic venous reflux in patients with pelvic varicose veins. J Vasc Surg Venous Lymphat Disord. 2021;9(6):1417-1424.

The authors position the manuscript as a review, so additional links to issue-related articles will help to increase the credibility and reliability of the statements presented in this work.

  1. It is reasonable to separate clinical and ultrasound diagnostics of acute and chronic diseases, accompanied by pelvic pain. This will improve the structure of the manuscript.

  1. In each subsection on the diagnosis of a particular disease, information should be added on the prevalence of this disease as a cause of pelvic pain.

  1. I would suggest to present not a separate “US diagnostic clue” for each disease, but a summary table at the end of the manuscript, presenting characteristics of pelvic pain and possibilities/limitations of ultrasonography for each of the described disorder.

  1. Pelvic pain has different characteristics in various diseases of the organs and veins of the pelvis. The authors describe clinical presentation and characteristics of pelvic pain for some diseases and not for others. Therefore, the common structure should be followed: the main features of the disease, clinical manifestations, ultrasound findings, and most importantly the features of the pain syndrome.

  1. Please add to the section “Pelvic congestion syndrome” that this disorder is characterized by not only the dilation of the ovarian veins with reflux in them (please also specify what reflux is considered pathological), but also by the dilation of the uterine and appendage (parametrial, uterine) veins with reflux in them. It is important to bear in mind that isolated ovarian vein incompetence is not the same as pelvic congestion syndrome (PCS). The latter one is characterized by the blood stasis in the veins of the uterus and appendages, which, in turn, occurs when there is reflux in the parametrial and uterine veins.

  1. It should be indicated that in clinical practice there may be thrombosis of not only the ovarian veins but also of the uterine and parametrial veins (in particular, in patients with pelvic varicose veins and patients after gynecological operations, radiation or hormone therapy). Thrombosis of these visceral veins is often asymptomatic and ultrasound data in such cases are decisive in the diagnosis.

  1. Given the number of diseases presented, the list of references is small for the literature review. More references should be added, including those above. This will make statements in the manuscript more convincing.

Author Response

Thank you for your comments helpful to improve manuscript quality

I am grateful to the editors of the Diagnostics journal for the opportunity to review the manuscript entitled: "Pelvic pain in reproductive age: US findings." The use of ultrasonography for identifying causes of pelvic pain is among the crucial issues not only in obstetrics and gynaecology, but also in vascular surgery, urology, and traumatology. In this regard, an important point is the standardization of the estimated parameters of abnormalities of pelvic organs and veins and retroperitoneal space. Unfortunately, an ultrasonographer is usually specialised in one area of medicine (e.g., in gynaecology or surgery), which leads to underdiagnosis of diseases outside his/her specialization. And underdiagnosis of pelvic venous disease in patients with pelvic pain is not such a rare example of that.

We fully agree. Our review article goes in this direction, also focusing the vascular causes of pelvic pain, often missed.

The presented manuscript confirms this thesis to a certain extent. Having chosen the most frequent organ causes of pelvic pain, the authors forgot about the methodological features of the study on the pelvic vessels (arteries and veins), while the correct interpretation of changes in pelvic veins can significantly increase the diagnostic value of ultrasonography in patients with pelvic pain.

What was recommended was explicated in the paragraphs dedicated to vascular pathology, specifying the correct examination technique.

One should not forget about visceral artery aneurysms, post-thrombotic lesions of the iliac veins.

We have considered exclusively causes of pelvic pain in fertile women, so iliac vein thrombosis has not been included not having a specific gender identity and commonly manifested in an advanced age, excluding congenital alterations.

 Comments

  1. Regarding the title, it is not clear why the authors focused on only the reproductive age. Pelvic pain may develop at any age. Moreover, the risks and incidences of diseases associated with pelvic pain in women increase with aging. In my opinion, the emphasis on age should be removed from the title.

We fully agree with the reviewer, however, we preferred to electively focus on this age range as the US is often the first and only diagnostic reference in this patients category, species for radiation protection issues. Therefore, we believe is fundamental in this age range to focus on the peculiar aspects of ultrasound imaging that can allow a clinical-ultrasound diagnosis. Furthermore, the pathologies treated, although can be found in advanced age, are more frequent in this age group.

The integration of the manuscript beyond this age range should include also the neoplastic conditions that are outside of the intent of our review article, as for this pathologies ultrasound,  although represent the first diagnostic step does not represent the conclusive imaging neither for diagnosis nor for stadiation purpose.

  1. Examination technique. I fully admit that the standard protocol for examining the pelvic organs does not require a detailed description. However, with regard to the examination of pelvic veins, it is necessary to describe features of the study of pelvic and retroperitoneal veins: the position of the patient (there are several of them), stress tests, veins classified as pelvic, the study of main venous trunks (left renal vein, iliac veins), and reflux.

In this regard, the authors should get acquainted with the following articles and cite them in the References: 

Whiteley MS, Dos Santos SJ, Harrison CC, Holdstock JM, Lopez AJ. Transvaginal duplex ultrasonography appears to be the gold standard investigation for the haemodynamic evaluation of pelvic venous reflux in the ovarian and internal iliac veins in women. Phlebology. 2015;30(10):706-713;

Gavrilov S, Karalkin A, Mishakina N, Efremova O, Grishenkova A. Relationships of Pelvic Vein Diameter and Reflux with Clinical Manifestations of Pelvic Venous Disorder. Diagnostics (Basel). 2022;12(1):145;

Gavrilov S, Moskalenko YP, Mishakina NY, Efremova OI, Kulikov VM, Grishenkova AS. Stratification of pelvic venous reflux in patients with pelvic varicose veins. J Vasc Surg Venous Lymphat Disord. 2021;9(6):1417-1424.

The authors position the manuscript as a review, so additional links to issue-related articles will help to increase the credibility and reliability of the statements presented in this work.

We thank the Reviewer for the suggestions that helped in improving the paragraph concerning pelvic congestion syndrome. The examination technique, the diagnostic references and the article references indicated have been widely integrated in the dedicated paragraph.

[...] In this regard, it appears critical to have a rigorous methodological US approach of pelvic and retroperitoneal veins combining TSA-US and TVS-US with dynamic color-Doppler Valsalva maneuvers performed in the patient’s supine, half-sitting (with trunk raised to 45°) and half-standing positions [40, 41]. TVS-US with color-Doppler is considered as the gold standard investigation for the hemodynamic assessment of pelvic veins reflux in women since it offers better visualization of the pelvic venous plexus compared to TSA-US, and is not hampered by patient habitués or undisplaceable bowel gas [37, 40,41,42]. TSA-US with colour Doppler of the iliac veins, inferior vena cava, renal and gonadal is useful in searching functional causes of pelvic reflux such as incompetent gonadal vein valves or structural causes of pelvic reflux such as renal vein, iliac vein and/or inferior vena cava compression and/or abnormalities too [40-42].  Finally duplex-ultrasonography of the veins of lower extremities is also a necessary part of the imaging protocol for improved evaluation of pelvic congestion syndrome, especially in the presence of atypical varicose veins [37].

  1. It is reasonable to separate clinical and ultrasound diagnostics of acute and chronic diseases, accompanied by pelvic pain. This will improve the structure of the manuscript.

We thank the Reviewer for the suggestions. Actually, we have preferred to structure the article in an anatomic-ultrasound order of pathologies, as adnexal, uterine and vascular causes. However, following the suggestion, a new table was added (Table 1) in which the epidemiological data and the most frequent clinical presentation (acute/chronic) were included for each pathology.

  1. In each subsection on the diagnosis of a particular disease, information should be added on the prevalence of this disease as a cause of pelvic pain.

The incidence/prevalence for each pathology has been reported in each paragraph as well as in Table 1.

  1. I would suggest to present not a separate “US diagnostic clue” for each disease, but a summary table at the end of the manuscript, presenting characteristics of pelvic pain and possibilities/limitations of ultrasonography for each of the described disorder.

Hoping that it can be acceptable, we preferred to add Table 1, in which the epidemiological data and the most frequent clinical presentation (acute/chronic) were included. In the remaining tables, US limits were added, to give a conceptual order for a quick reading, otherwise we would have a single table too loaded with concepts.

  1. Pelvic pain has different characteristics in various diseases of the organs and veins of the pelvis. The authors describe clinical presentation and characteristics of pelvic pain for some diseases and not for others. Therefore, the common structure should be followed: the main features of the disease, clinical manifestations, ultrasound findings, and most importantly the features of the pain syndrome.

In each paragraph this order has been carefully kept and in the Table 1 were underlined the clinical data.

  1. Please add to the section “Pelvic congestion syndrome” that this disorder is characterized by not only the dilation of the ovarian veins with reflux in them (please also specify what reflux is considered pathological), but also by the dilation of the uterine and appendage (parametrial, uterine) veins with reflux in them. It is important to bear in mind that isolated ovarian vein incompetence is not the same as pelvic congestion syndrome (PCS). The latter one is characterized by the blood stasis in the veins of the uterus and appendages, which, in turn, occurs when there is reflux in the parametrial and uterine veins

We thank the Reviewer for this valuable advice. We pointed out this aspect in this paragraph:

[ ...]the diameter of pelvic veins is not a diagnostic criterion to differentiate symptomatic and asymptomatic PCS, how much more the duration of pelvic venous reflux greater than 1second, its prevalence in the pelvic veins and blood deposition in the pelvic venous plexuses, including the uterine and parametrial veins and not only the ovarian ones, are usually the leading factors in the development of symptomatic forms of pelvic congestion (Table 12) [40, 41].

  1. It should be indicated that in clinical practice there may be thrombosis of not only the ovarian veins but also of the uterine and parametrial veins (in particular, in patients with pelvic varicose veins and patients after gynecological operations, radiation or hormone therapy). Thrombosis of these visceral veins is often asymptomatic and ultrasound data in such cases are decisive in the diagnosis.

We thank the Reviewer for this valuable advice. We pointed out this aspect in this paragraph:

Uterine venous plexus thrombosis is also an unusually site of thrombosis often asymptomatic and incidental detected by TSV-US which is decisive in the diagnosis and subsequent therapeutic choices [45].

  1. Given the number of diseases presented, the list of references is small for the literature review. More references should be added, including those above. This will make statements in the manuscript more convincing.

The reference list was widely extended, adding the following references :

3,710,11,12,13,19,22,27,28,29,30 34,35, 37,38,39,40,41,42, 44,45,46,47

Round 2

Reviewer 2 Report

Table 1 should be removed from the "Introduction" section and placed in the "Examination technique" section with an appropriate comment.

There are no other comments.

Author Response

Table 1 should be removed from the "Introduction" section and placed in the "Examination technique" section with an appropriate comment.   We thank the Reviewer for the suggestion. Table 1 was renamed  table 2 changing the reference paragraph as suggested.